

# Research progress on the impact of climate change on wheat production in China

Yu-chen Fan[1,*], Ya-qi Yuan[1,*], Ya-chao Yuan[1], Wen-jing Duan[2] and Zhi-qiang Gao[1]

[1] College of Agriculture, Shanxi Agricultural University, Shanxi, Jinzhong, China
[2] Department of Biological and Technology, Jinzhong University, Shanxi, Jinzhong, China
* These authors contributed equally to this work.

## ABSTRACT

It is crucial to elucidate the impact of climate change on wheat production in China. This article provides a review of the current climate change scenario and its effects on wheat cultivation in China, along with an examination of potential future impacts and possible response strategies. Against the backdrop of climate change, several key trends emerge: increasing temperature during the wheat growing season, raising precipitation, elevated $CO_2$ concentration, and diminished radiation. Agricultural disasters primarily stem from oscillations in temperature and precipitation, with the northern wheat region being mostly affected. The impact on wheat production is manifested in a reduction in the area under cultivation, with the most rapid reduction in spring wheat, and a shift in the center of cultivation to the west. Furthermore, climate change accelerates the nutritional stage and shortens phenology. Climate change has also led to an increase in yields in the Northeast spring wheat region, the Northern spring wheat region, the Northwest spring wheat region, and the North China winter wheat region, and a decrease in yields in the middle and lower reaches of the Yangtze River winter wheat region, the Southwest winter wheat region, and the South China winter wheat region. To cope with climate change, Chinese wheat can adopt adaptation strategies and measures such as breeding different wheat varieties for different wheat-growing regions, implementing differentiated farmland management measures, promoting regional ecological farmland construction, and establishing scientific monitoring and early warning systems. While future climate change may stimulate wheat yield potential, it could cause climate-induced issues such as weeds, diseases, and pests worsen, thereby posing challenges to the sustainability of farmland. Moreover, it is essential to conduct comprehensive research on pivotal areas such as the microscopic mechanism of climate change and wheat growth, the comprehensive influence of multiple climate factors, and the application of new monitoring and simulation technologies. This will facilitate the advancement of related research and provide invaluable insights.

## INTRODUCTION

The ongoing global warming trend persists, with the period between 2015 and 2022 registering as the warmest 8 years within the span of the 19th and 20th centuries. Among

Corresponding authors
Wen-jing Duan,
duanwenjings@163.com
Zhi-qiang Gao, gaosxau@163.com

them, China has distinct warm-dry climate characteristics and is a sensitive area and an area with significant influence of global climate change. Moreover, China is also one of the world's largest greenhouse gas-emitting countries. Therefore, China's $CO_2$ concentration, air temperature, precipitation, and extreme weather changes are of great significance for the global climate change assessment. According to the Sixth Assessment Report of the Intergovernmental Panel on Climate Change (IPCC), human-emitted greenhouse gases such as $CO_2$ contribute 76% to climate change (*Gray, 2007*). According to the data released by the International Energy Agency (IEA), in 2023, China's $CO_2$ emissions increased by approximately 565 million tons, being the country with the largest increase. Meanwhile, China's annual average surface temperature has been rising at a rate of 0.26 °C/10 years (1951–2020), an increase of about 1.8 °C from 1951 to 2020, and an increase of 0.92 °C from 2021 to 2022 (*National Climate Center, 2023*). The rise in temperature is accompanied by the melting of frozen ground and rising sea levels, thereby augmenting the propensity for compounded extreme weather occurrences, resulting in substantial regional rainfall and ecological aridity (*Peterson et al., 2008*). The year 2021 saw a delayed and subdued onset of the rainy season in the southern regions of China, in contrast to an early and intensified manifestation in the northern territories. The daily rainfall in Zhengzhou, Henan Province, reached 552.5 mm, breaking the record of the daily rainfall since 1951 and approaching the average annual rainfall of 640.8 mm in Zhengzhou. The frequency of extreme high temperature events in China has shown a significant increasing trend (1961–2022), with 2022 registering the highest frequency of such events since 1961, increasing by 7.3 days compared with the perennial average, with an increase rate of 80.2%. Therefore, China needs to face climate challenges including heightened $CO_2$ concentrations, temperature elevation, intensified precipitation, and ecological aridity, *etc*.

The relationship between climatic factors and wheat production is close and complex, and the yield is comprehensively affected by climate change. The crop model Agricultural Production Systems Simulator (APSIM) evaluated that when the $CO_2$ concentration in Indian wheat increased from 350 to 550 ppm, the wheat biomass increased by 35% and the yield increased by 33% (*Mohanty et al., 2015*). However, it can also weaken nitrogen metabolism, leading to a decline in the nutritional quality of grains and the enrichment of trace elements such as zinc (*Högy & Fangmeier, 2008*). Rising temperatures will reduce the growth time of wheat. For every 1 °C increase in temperature, the yield will be reduced by 5–6% (*Jacott & Boden, 2020*). However, for the Qinghai-Tibet Plateau region at high altitudes, rising temperatures will increase the effective accumulated temperature, which increases by 16.2–173.3 °C·d per decade (*Zhang et al., 2023*), thus promoting the increase in yield. Precipitation can affect the soil moisture in wheat fields and improve soil structure and fertility (*Reed et al., 2012*), especially having a significant impact on drylands. However, heavy rainfall can also cause flood disasters. For example, in the rice–wheat area of the Somb Basin in Haryana, India, a minor flood occurs once every 2–3 years and a major flood occurs at least once every 10 years, flooding farmlands (*Singh & Singh, 2015*), causing anoxia in crop roots, poor plant growth, and resulting in a reduction in yield. Insufficient precipitation increases the input of farmland irrigation and reduces the economic benefits of wheat (*Lu et al., 2019*). The occurrence of extreme weather will lead

to a decline in wheat yield. For example, from 1991–2014 different wheat varieties in Europe almost all showed a reduction in yield when facing extreme climates (high temperature, drought, heavy rainfall) (*Mäkinen et al., 2018*). Therefore, the impact of climate change on wheat needs to be comprehensively evaluated, and the interaction between climatic factors and regional differences need to be thoroughly considered.

Wheat is one of China's most important staple crops. According to the statistics of the National Bureau of Statistics of China, the second largest planted area after maize and rice in 2019 and a total yield of $1.3360 \times 10^8$ t, accounting for 20.1% of total cereal production (*Huang et al., 2017*). Meanwhile, the total wheat output in China has maintained a slow-growth trend. However, China is facing the predicament of a reduction in wheat-planting area. Statistical data reveal a discernible contraction in cultivated acreage, witnessing a reduction of $9.38 \times 10^5$ hectares between 2016 and 2019 (*Huang et al., 2017*; *Zhang et al., 2023*). Therefore, the diminishing cultivated area emerges as a pivotal constraining factor influencing aggregate wheat production (*Zhao, Yang & Sun, 2018*). Breaking the wheat mono-yield barrier to achieve ultra-high yields is crucial to ensure wheat food security. Moreover, against the background of global climate change, the pursuit of enhanced yield thresholds encounters heightened challenges (*Dan et al., 2024*; *Zhang, Niu & Yu, 2021*). Thus, elucidating the contemporary and prospective ramifications of climate change on indigenous wheat cultivation ecosystems and yields assumes pivotal significance, and proposing adaptive strategies to deal with the challenges of climate change. Such attempts not only provide a theoretical underpinning for realizing ultra-high wheat productivity, but also furnish cogent insights for climate change appraisal frameworks in China and even around the world.

## SURVEY METHODOLOGY

The primary retrieval tools used for our research are Google Scholar and the China National Knowledge Infrastructure (CNKI) database, with a few Chinese articles being retrieved from the latter. The initial screening criteria for articles are designed to guarantee the precision of search terms. The search terms for this article were "China wheat," "climate," "temperature," "carbon dioxide," "radiation," "yield," "growth and development." Secondly, select high-impact or highly-cited journals and articles, as well as time-sensitive literature from the last 5–10 years. In addition, we have selected some highly cited but older publications as theoretical foundations or references. Furthermore, select articles with high accuracy and a long research duration to ensure that the research conclusions are highly generalizable and summarizable. Finally, select articles with diverse perspectives. The articles should include both the two-sided impacts of climate change on Chinese wheat and regional diversity, ensuring that all wheat-growing regions in China are covered as much as possible. Moreover, corresponding adaptive wheat-field management measures are proposed specifically to cope with the negative impacts of climate, and the biological mechanisms of the impacts of climate change on Chinese wheat are analyzed. Additionally, given the abundance of wheat research within China, we utilized the CNKI database, the largest Chinese-language database, to select a portion of the literature.
However, it should be noted that these selected articles are globally accessible and constitute only a small proportion of the overall review.

However, there are still certain limitations in our research. In this article, only a few databases were selected, namely Google Scholar and CNKI. Compared with Web of Science, there are deficiencies in citation tracking and research network, and some articles with low influence or low citation may be missed. Moreover, there are language limitations in the retrieval. We carried out the retrieval in English and Chinese, lacking non-English and non-Chinese articles. Therefore, the richness of our research is restricted. In future research, we need to gradually learn to use EBSCOhost to retrieve in multiple languages, compare multiple databases such as Google Scholar and PubMed, and conduct reverse retrieval (finding other articles that cite a certain article). Also, we should pay attention to low-impact journals to improve publication bias and citation bias (*Olsson & Sundell, 2023*). It may take our team a long time to do this. At the same time, there are also imperfections and inaccuracies in the search terms in this article. There are differences in the translation of different regional names in China into English. We mainly used "China" as the search term, and less often used the names of wheat-growing regions as search terms, which may lead to some articles being missed. This serves as a warning for us. In future research, we need to strengthen communication with peers, collect local characteristic names, and learn the iterative retrieval method for optimizing search terms.

Furthermore, there are more studies on high-yield or large-planted-area wheat-growing regions, resulting in less content related to low-yield or small-planted-area wheat-growing regions in this article, which affects the integrity of the article. In future research and data collection processes, we need to pay more attention and devote more energy to low-yield or small-planted-area wheat-growing regions. Long-term data can be considered for simulation and prediction to provide theoretical references as much as possible and maintain the integrity and comprehensiveness of Chinese wheat research. Meanwhile, there also exists the time lag in publication (*Olsson & Sundell, 2023*). For example, published articles may use previous data, which is an objective limitation of data lag. In addition, writing articles is limited by our own knowledge background and research experience, which may lead to differences in the interpretation of the article. We need to keep strengthening learning and organizing to ensure the objectivity of the article as much as possible. Moreover, we mainly wrote the article by summarizing and commenting, without using statistical analysis, crop growth models, GIS technology, *etc.* to discuss the article in depth. Besides objective reasons such as differences in article research design, regional differences, and lack of unified standards for data collection in different regions in China, this also exposes our deficiencies and lack in handling data heterogeneity such as data standardization and conversion, and subgroup analysis. We need to participate in specific training, self-study, and practice for growth. Therefore, in future research, we will try our best to overcome these limitations. We will use multi-database retrieval, optimize search terms, multi-language retrieval, pay attention to low-impact journals, *etc.* to obtain the latest research status. Also, through multi-person review, clarifying the article framework, *etc.*, we will reduce the influence of subjectivity on the article. Moreover, we

will strengthen secondary data analysis, dig deeper into data, and promote the emergence of new viewpoints and the formation of new interpretation perspectives.

# EFFECTS OF CLIMATE CHANGE ON THE WHEAT GROWING ENVIRONMENT

## Characteristics of climate change

Greenhouse gas emissions have accelerated the rise in global temperatures (*Valone, 2021*). Using crop models (DSSAT-CERES-Wheat, DSSAT-Nwheat, WheatGrow and APSIM-Wheat), it was comprehensively assessed that for global temperatures rise by 1.5 °C, the average temperature during the winter wheat growth period will increase by 0.6–1.4 °C; if temperatures rise by 2.0 °C, the average temperature during the growth period will increase by 0.9–1.8 °C. For different wheat areas, the temperature increase is greater in spring wheat areas than in winter wheat areas, with the southwest wheat area showing a greater difference than the northern wheat area (*Ye et al., 2020*; *Wang, Zhan & Zou, 2023*; *Yang et al., 2023*). Among the winter wheat areas, Xinjiang emerges as the locus of the most substantial temperature rise, juxtaposed with the northwestern winter wheat areas recording comparatively modest increases. Notably, the Southwest winter wheat area and the Huang-Huai winter wheat area exhibit the largest and smallest temperature changes within the winter wheat ambit, respectively (*Sun et al., 2018*). The smallest winter wheat area has an annual temperature increase of 0.43, 0.35 and 0.54 °C for average, maximum and minimum temperature, respectively (*Chen et al., 2018*; *Sun, Wang & Wang, 2024*). In the spring wheat area, discernible escalations are observed in both average and minimum temperatures, with no significant interannual difference in the highest temperature, especially from the flowering to the ripening period (*Ye et al., 2021*). It is evident that climate change has increased the average temperature in domestic wheat areas, with differences between spring and winter wheat areas, and a greater impact on spring wheat areas than on winter wheat areas.

Climate change can lead to heavy regional precipitation and ecological drought (*Peterson et al., 2008*). In China, relative precipitation levels during the wheat growing season (2006–2010) exhibited a notable increase of 9.1–11.3% compared to the historical period spanning from 1986 to 2005 (*Sun et al., 2018*). However, discernible trends indicate a slight decrement in annual precipitation across the central and eastern regions of China (1961–2017) (*Song et al., 2019*). For wheat areas, the increase in precipitation is slightly greater in spring wheat areas than in winter wheat areas. Except for the Xinjiang spring wheat area, all spring wheat regions observed heightened precipitation levels, with the northern spring wheat area registering the most substantial increment. In the winter wheat area, with the exception of the northern winter wheat area and select locales in the central and western regions, an overall uptick in precipitation was observed, with the largest increase in the Huang-Huai winter wheat area (*Sun et al., 2018*; *Nie et al., 2019*). Evidently, climate change has engendered spatially and temporally changes in precipitation, with spring wheat territories witnessing a comparatively greater increase in precipitation than their winter wheat counterparts.

Climate change also affects factors such as solar radiation and $CO_2$ concentrations (*Peterson et al., 2008*; *Gray, 2007*). Due to the large amount of pollutants emitted into the air by human activities (*Xiong et al., 2012*), solar radiation in China decreased by about 0.23 W/(m$^2$·yr) from 1960 to 2000 (*Tang et al., 2011*). China's atmospheric $CO_2$ concentration was about 365 ppm in 2000 (*Deng et al., 2020*), and it exceeded 419 ppm in 2022 (the data is from the *China Meteorological Administration, 2023*), an increase of 14.8%. This dual trend of rising $CO_2$ concentrations alongside diminished solar radiation underscores the impact of climate change on these pivotal environmental parameters.

In summary, the ramifications of climate change extend to an elevation in average temperatures across domestic wheat-growing regions, an erratic spatial and temporal precipitation pattern, augmented $CO_2$ concentrations, reduced solar radiation, and a disproportionately greater impact on spring wheat territories. These climate factors, in addition to causing meteorological disasters, can also jointly affect the entire growth process and yield formation of wheat, which is very worrying. How to respond to climate change based on local conditions will be a huge challenge for wheat in China.

## Characteristics of agricultural meteorological disasters

There are many agricultural meteorological disasters affecting wheat, including drought, hail, heavy rain, high temperature and frost damage, with drought and frost damage being the most important meteorological disasters (*Gray, 2007*; *Ye et al., 2020*; *Zou et al., 2021*). Between 1991 and 2009, agricultural meteorological observation stations documented 5,902 instances of wheat-related disasters, with natural disasters due to abnormal precipitation being the most common, followed by temperature. High temperatures and low rainfall leading to drought accounted for 79.2% of the total disaster occurrences, while excessive precipitation contributed to 4.0%, and frost damage accounted for 1.32% (*Zhang et al., 2013*). On average, approximately $4.3 \times 10^7$ hectares of land are affected annually by these natural disasters. Notably, the collective impact of floods, droughts, and frosts covers an average annual area of $3.6 \times 10^7$ hectares (*Jiang & Cui, 2016*). These observations underscore the primary role of precipitation and temperature fluctuations in precipitating agricultural meteorological disasters across the nation.

In terms of regions, the Northern Wheat Region is a meteorological disaster-prone area, with Gansu being the most affected province (*Zhang et al., 2013*), with the most affected provinces being Heilongjiang, Shandong and Henan, and the least disaster resilient provinces being Shanxi and Inner Mongolia (*Yu et al., 2017*). The spatial pattern of disasters from 1950 to 2013 shows that there are more flood disasters in the middle and lower reaches of the Yellow River and the Yangtze River basin, while drought is more common in the North China Plain and the Loess Plateau (*Zhang et al., 2013*). The central and south-western regions have more drought (*Guan et al., 2015*), while the north-western region has more frost damage and the north-eastern region has more drought (*Yu et al., 2017*). This spatial disparity in meteorological disasters highlights the northern region as a high-risk area, delineating significant regional variations in vulnerability.

The frequency of disasters in the wheat reproductive stage is higher than in the nutritional stage (*Hussain et al., 2016*). From 1991 to 2009, the disaster rate before jointing

was 25.1%, which increased to 74.9% after jointing. Among the disaster types, 72% of drought, 88.1% of heavy rain and 100% of high temperature disasters occurred during the reproductive stage. Therefore, anomalies in atmospheric circulation resulting in uneven rainfall distribution (*Jiang & Cui, 2016*) and frequent extreme temperatures (*Guan et al., 2015*), as well as the spatial diversity of disasters, are more prevalent in the reproductive stage. This situation calls for strengthening the resilience and adaptability of wheat varieties, alongside the implementation of innovative field management strategies during the later stages of growth.

In summary, the meteorological disasters suffered by Chinese wheat mainly come from drought and flood caused by precipitation, high temperature and freezing damage caused by temperature. Regionally, the disaster-affected areas in the northern wheat-growing regions are more than those in the southern regions, and the reproductive stage is more affected than the vegetative stage.

## EFFECTS OF CLIMATE CHANGE ON WHEAT PRODUCTION

### Impact on wheat phenology

Both winter and spring wheat, as well as different wheat regions, respond differently to climate change, resulting in differences in phenology (*Valone, 2021*; *Ye et al., 2020*; *Inouye, 2022*; *Wang et al., 2022*). Adequate water can support physiological processes such as cell division, elongation, and differentiation in wheat phenology, which makes the phenology of winter wheat and spring wheat increase with the increase in precipitation. Among them, the root system of winter wheat is relatively shallow, and it experiences three seasons: autumn, winter, and spring. Therefore, winter wheat is more sensitive to precipitation. An increase in temperature will accelerate the low-temperature vernalization and physiological metabolism speed of winter wheat. Moreover, the increase in temperature increases the risk of water stress in winter wheat, resulting in the decrease of winter wheat phenology with the increase in average temperature, while spring wheat is the opposite (*Wang et al., 2022*). This is because the increase in temperature weakens the low-temperature stress on spring wheat during the seed germination stage, enabling it to start the growth process normally. However, the spatio-temporal non-uniformity of precipitation can be dealt with through management methods such as improving irrigation facilities and methods (ensuring soil moisture, for example, using drip irrigation and soil-moisture-measured supplementary irrigation in North China), establishing rainwater harvesting systems (collecting rainwater during periods of abundant precipitation for irrigation during drought periods), and tillage and mulching for soil moisture conservation (combining tillage methods such as deep tillage, shallow tillage, and no-tillage with mulching to enhance soil water-holding capacity and reduce soil water evaporation) (*Zhang et al., 2023*). The increase in temperature can be coped with through management methods such as variety improvement (selecting heat-tolerant varieties for winter wheat and varieties with strong temperature adaptability for spring wheat), adjusting the sowing date (delaying the sowing date of winter wheat to deal with warm winters, and sowing spring wheat earlier to ensure the accumulation of organic matter in the later growth stage), reducing the planting density (improving the ventilation and light-transmission

conditions in the wheat field to reduce competition among plants), and increasing the irrigation frequency and time (conducting sprinkler irrigation or micro-irrigation before the temperature rises to reduce the field temperature through water evaporation) (*Ye et al., 2020*; *Chen et al., 2018*; *Nie et al., 2019*; *He et al., 2019*; *Zhang et al., 2023*).

The response of wheat regions to climate is reflected in area and center of gravity, with significant effects of precipitation and temperature on area, contributing 11.1–13.1% and 9.7–14.1% respectively (*Fan et al., 2018*; *Tao et al., 2014*; *Liu et al., 2018*). Temperature has a significant effect on the center of gravity, with a positive drive for winter wheat and the opposite for spring wheat. From 1949 to 2014, the planting area experienced a turning point in the 1970s, with a decrease rate of $0.89 \times 104$ and $1.99 \times 10^4$ ha·a$^{-1}$ for winter wheat, and an increase rate of $0.27 \times 10^4$ ha·a$^{-1}$ followed by a decrease rate of $8.46 \times 10^4$ ha·a$^{-1}$ for spring wheat. The center of gravity of planting shifted 31 km from Henan to the northwest for winter wheat and 692 km from Inner Mongolia to the southwest for spring wheat (*Valone, 2021*; *Liu et al., 2018*). In response to the reduction in the planting area and the westward shift of the planting center of gravity, the method of increasing the yield per unit area can be adopted. Specific measures include high-quality varieties (high-photosynthetic-efficiency, high-yield, and strong-stress-resistance varieties), reasonable soil structure (a combination of deep plowing, no-tillage, and straw mulching, which not only breaks the plow pan and enhances soil aeration but also reduces soil disturbance and maintains a good soil microbial environment), rational close-planting (for varieties with a compact plant type and weak tillering ability, the planting density can be increased; for varieties with a loose plant type and strong tillering ability, the planting density can be appropriately reduced), precise irrigation (according to the water-requirement law of wheat), and pest and disease monitoring and prevention, *etc.* (*Ye et al., 2020*; *Chen et al., 2018*; *Nie et al., 2019*; *He et al., 2019*; *Adedeji et al., 2020*; *Zhang et al., 2023*).

Climate change accelerates the physiological metabolism through temperature increase, promotes or stresses due to the spatio-temporal non-uniformity of water, and increases the photosynthetic efficiency due to the elevated carbon dioxide concentration. Moreover, the shortening of the vernalization stage of winter wheat and the smooth start of spring wheat seed germination accelerate the growth process of wheat (*Zhao, Yang & Sun, 2018*; *Yang et al., 2023*; *Sun et al., 2018*). From 1981 to 2010, climate change led to earlier of both the flowering and maturity of different wheat varieties (*Inouye, 2022*; *Liu et al., 2018*), resulting in a 0.23 days per year shortening of the nutritional stage and 0.06 days per year lengthening of the reproductive stage (*Wang et al., 2022*; *Liu et al., 2018*). In 2010, the nutritional stage was shortened by 6.9 days and the reproductive stage was lengthened by 1.8 days. Different growth stages of wheat were affected differently: the sowing, emergence and three-leaf stages were delayed by 0.19 days per year, 0.06 days per year and 0.05 days per year respectively, while the tillering, jointing, booting, heading, flowering and ripening stages were advanced by 0.02 days per year, 0.15 days per year, 0.17 days per year, 0.19 days per year and 0.10 days per year respectively (*Wang et al., 2022*; *Liu et al., 2018*). In response to the acceleration of the vegetative stage caused by climate change, varieties with a long vegetative period, multiple tillers and well-developed root systems can be adopted, so that the wheat roots can absorb more nutrients, tillers can directly increase the photosynthetic

area, and sufficient vegetative growth time can be maintained. Sowing can also be postponed to slow down the speed at the initial stage of vegetative growth and extend the vegetative growth time (*Ye et al., 2020*; *Akman, Yildirim & Bağci, 2023*).

In summary, climate change has led to a reduction in area and a westward shift in the center of gravity, with a much higher rate of reduction in spring wheat area than in winter wheat. This has led to uncertainty about investment in agricultural infrastructure. The shortened phenology and accelerated growth stages pose new challenges for wheat variety breeding and field management.

## Impact on the wheat management system

Adaptive management of wheat fields can mitigate the negative effects of climate change (*Dan et al., 2024*; *Valone, 2021*; *Ye et al., 2021*). Field irrigation can increase the temperature tolerance of wheat (*Liu, Zhang & Ge, 2021*). Irrigation reduces the stress of high temperatures during wheat growth and improves water use efficiency. Water use efficiency in the central region increased by 34.2%, and with rising temperatures, water use efficiency also increased, especially in dry years (*Wu et al., 2020*). Spring wheat uses redundant growth during the nutritional stage to cope with temperature increases during the reproductive stage. Controlling the timing and amount of irrigation can reduce this redundancy and also help to alleviate the stress of high temperatures (*Bai, Du & Miao, 2021*). In the Hetao region, delaying the heading stage by 15 days and applying 50–70 mm of irrigation resulted in increased yields and incomes (*Bai, Du & Miao, 2021*). The sowing date can alter the accumulated temperature of wheat, thereby regulating its growth and development (*Dreccer et al., 2013*; *Shang et al., 2023*). Late sowing of winter wheat reduces the pre-winter accumulated temperature, delays tillering during the seedling stage and consequently delays flowering, with the optimum late sowing being 5–7 days (*Ren et al., 2019*).

Early sowing of spring wheat can extend the growth period of wheat (nutritional stage, reproductive stage) and reduce the average temperature during the growth stages to cope with rising temperatures (*Xiao et al., 2016*; *Li & Lei, 2022*). In addition, greenhouse gas emissions can be reduced through the timing and amount of fertiliser application, combined application of organic and inorganic fertilisers and other methods (*Lyu et al., 2019*; *Li et al., 2023a*), while soil management practices can reduce carbon footprints and mitigate climate change impacts (*Zhang et al., 2021*; *Shang et al., 2023*). Therefore, it is evident that field management practices such as irrigation, sowing dates, fertilisation and tillage can adjust wheat growth and development or greenhouse gas emissions and thus serve the purpose of addressing climate change.

## Effects on wheat grain yield

The wheat yield in China varies with the changes of climatic factors such as temperature, precipitation, $CO_2$, and radiation (*Peterson et al., 2008*; *Gray, 2007*; *Zhao, Yang & Sun, 2018*; *Zhang et al., 2021*; *Lyu et al., 2019*; *Li et al., 2023b, 2023c*). From 1981 to 2009, wheat yields in the northern regions exhibited an increment ranging from 0.9% to 12.9%, juxtaposed with a decrease ranging from 1.2% to 10.2% observed in the southern regions
(*Tao et al., 2014*). An increase in temperature is beneficial for the accumulation of organic matter during the grain filling period, leading to an increase in yield (*He et al., 2020*). Specifically, a temperature elevation of 1.5 °C corresponds to a yield increase of 2.8%, while a 2.0 °C rise yields an 8.3% increment (*Wang, Zhan & Zou, 2023*). Wheat yield variability is influenced by rainfall in May, with 70–78% of the variability attributed to precipitation. Excessive precipitation weakens photosynthesis during the grain filling period and can induce Fusarium head blight, which reduces grain quality (*Song et al., 2019*). It is evident that precipitation exerts a significant impact on both flowering and grain-filling phases, with diminished precipitation correlating with reduced yields (*Yu et al., 2018*; *Geng et al., 2023*). Reduced radiation contribute to yield reduction, albeit partially mitigated by increased $CO_2$ concentrations (*Xiong et al., 2012*).

The increase in temperature promotes the physiological metabolic activities of wheat plants, especially the enhancement of photosynthesis, which can synthesize and accumulate more organic matter, thus increasing wheat yield. Precipitation significantly affects wheat yield, among which the precipitation in May has the greatest impact, causing 70–78% of the variation in wheat yield. May is a crucial period for wheat yield formation. Excessive precipitation will weaken photosynthesis during the filling period and induce Fusarium head blight, reducing grain quality (*Song et al., 2019*). Too little precipitation will cause the closure of leaf stomata, blocked carbon dioxide intake, reduced activity of photosynthetic pigments and enzymes, decreased photosynthetic reaction efficiency, and weakened water transport and transpiration pull in wheat, resulting in limited transport of photosynthetic products, poor grain development, and finally a reduction in wheat yield (*Yu et al., 2018*; *Zhao et al., 2020*). Moreover, before winter, wheat seedlings have a low demand for water and a certain degree of drought resistance. The intensity of physiological processes (such as photosynthesis and material transport) involving water is low. In addition, the temperature before winter is relatively low, and the transpiration of wheat is weak, so water loss is small (*Yu et al., 2018*). Therefore, an appropriate amount of effective precipitation can have a positive impact on wheat yield. $CO_2$ concentration and radiation are respectively the raw material and energy source for photosynthesis in wheat leaves, and they affect wheat yield through the synthesis of organic matter. The increase in $CO_2$ concentration absorbs long-wave radiation, resulting in a decrease in ground radiation. That is, the increase in $CO_2$ concentration promotes photosynthetic synthesis, while the decrease in radiation restricts the progress of photosynthesis. Interestingly, the yield increase due to the rise in $CO_2$ concentration can just compensate for the yield reduction caused by the decrease in radiation (*Xiong et al., 2012*). Moreover, the combined effect of climatic factors on wheat yield in China may vary by region. Analyzing China's wheat yield from 1981–2009, it was found that the wheat yield in the northern region increased by 0.9–12.9%, while that in the southern region decreased by 1.2–10.2% (*Tao et al., 2014*). Conclusively, the multifaceted effects of climate change manifest both positive and negative repercussions on yield dynamics, culminating in augmented production in northern regions juxtaposed with diminished output in southern regions.

The response of China's wheat yield to climate change varies depending on the wheat-growing areas and wheat types (Table 1). The increase in temperature caused by climate change has compensated for the key factor of insufficient temperature for wheat. China's relatively complete farmland irrigation and water-collecting facilities have reduced the risk of wheat in dealing with the spatio-temporal non-uniformity of precipitation or drought (*Chen et al., 2018*; *Zhu & Song, 2021*). Therefore, the rising temperature promotes the spring wheat in the northeast spring wheat area, the northern spring wheat area and the northwest spring wheat area. In the early growth stage, it is beneficial for tiller formation, increasing the number of spikes (*Ye et al., 2021*). In the later growth stage, it reduces sterile spikelets and increases the number of grains per spike. The rising temperature has increased the effective accumulated temperature and organic matter accumulation of winter wheat in the North China winter wheat area, promoting a 2.1% increase in winter wheat yield. There is a compensatory effect between light and $CO_2$ concentration, while pests and diseases are affected by climate factors such as high temperature, flood and drought, which brings pressure on the prevention and control of pests and diseases and wheat breeding. Although measures such as constructing a plant protection prevention and control system for pests and diseases, tillage measures and rational close-planting can be taken, the yields in the Middle and Lower Reaches of the Yangtze River winter wheat area, the Southwest winter wheat area and the South China winter wheat area have decreased. Moreover, the response of southern winter wheat yield to temperature in this century is different from that in the last century (*Wang et al., 2022*; *Fan et al., 2018*; *Tao et al., 2014*; *Liu et al., 2018*; *Ren et al., 2019*; *Wang et al., 2023a*; *Xiao et al., 2016*; *Li & Lei, 2022*). In the last century, an increase in temperature promoted an increase in wheat yield, with an increase of 1.4–9.6%; in this century, the south decreased with the increase in temperature (*Wang et al., 2022*), with an average temperature increase of 1 °C leading to a yield decrease of 4.0% in the south (*Chen et al., 2018*). The decrease in yield mainly results from the intensification of pests and diseases and the frequent occurrence of extreme rainfall. Moreover, the Southwest winter wheat region and the South China winter wheat region are respectively faced with the challenges of scattered arable land and small planting area. Evidently, the escalating temperatures have engendered divergent impacts on winter and spring wheat yields across geographical regions, winter wheat and spring wheat yields have increased in the north, while winter wheat yields have decreased in the south. In summary, climate change in the last century favoured an increase in the yield of winter (spring) wheat. In this century, the impact of climate change has resulted in amplified production of northern winter (spring) wheat while concurrently precipitating diminished production of southern winter wheat (*Xiao et al., 2016*). However, it is imperative to acknowledge that winter wheat encompasses a substantial majority, which accounts for about 94% of China's aggregate wheat output (*Huang et al., 2017*). Consequently, the contemporary challenge lies in the pursuit of attaining heightened yields in northern wheat territories alongside the imperative of ensuring stable yields in southern winter wheat regions.

**Table 1 Key factors for yield formation in different wheat-growing areas and wheat types.**

| Wheat-growing areas | Wheat types | The response of yield to climate change and key factors | | |
| --- | --- | --- | --- | --- |
| | | Response situation | Key factors | Sub-key factors |
| Northeast Spring Wheat Area | Spring wheat | Yield increase (*Ye et al., 2020*; *Wang et al., 2022*) | Insufficient temperature (*Valone, 2021*) | Spatio-temporal non-uniformity of precipitation (*Yu et al., 2018*) |
| Northern Spring Wheat Area | Spring wheat | Yield increase (*Ye et al., 2020*; *Wang et al., 2022*) | Insufficient temperature (*Valone, 2021*) | Spatio-temporal non-uniformity of precipitation (*Yu et al., 2018*) |
| Northwest Spring Wheat Area | Spring wheat | Yield increase (*Nie et al., 2019*; *Wang et al., 2022*) | Drought (*Nie et al., 2019*) | Insufficient temperature (*Valone, 2021*; *Nie et al., 2019*) |
| North China Winter Wheat Area | Winter wheat | Yield increase (*Wang et al., 2022*; *Zhang et al., 2023*) | Insufficient temperature (*Valone, 2021*) | Spatio-temporal non-uniformity of precipitation (*Yu et al., 2018*) |
| Middle and Lower Reaches of the Yangtze River Winter Wheat Area | Winter wheat | Yield increase (*Wang et al., 2022*; *Zhang et al., 2023*) | Insufficient sunlight (*Xiong et al., 2012*) | Frequent pests and diseases (Perry et al., 2012) |
| Southwest Winter Wheat Area | Winter wheat | Yield decrease (*Ye et al., 2020*; *Wang et al., 2022*) | Insufficient sunlight (*Xiong et al., 2012*) | Frequent pests and diseases, scattered arable land (*Ye et al., 2020*; Perry et al., 2012) |
| South China Winter Wheat Area | Winter wheat | Yield decrease (*Wang et al., 2022*; *Liu et al., 2020*) | Insufficient sunlight (*Xiong et al., 2012*) | Frequent pests and diseases, small planting area (Perry et al., 2012; *Liu et al., 2020*) |

# ADAPTIVE STRATEGIES AND MEASURES FOR CHINESE WHEAT IN RESPONSE TO CLIMATE CHANGE

Future climate change is expected to be more severe, characterized by sustained elevations in average temperature and $CO_2$ concentration, alongside exacerbated spatial and temporal variability in precipitation patterns (*National Climate Center, 2023*; *Peterson et al., 2008*; *Gray, 2007*; *Ye et al., 2021*; *Tao et al., 2014*; *Qin et al., 2015*; *Prodhan et al., 2022*). The Chinese wheat industry has been confronted with a series of challenges and opportunities. These include the impact of hot, dry winds (*Chen et al., 2016*), floods and droughts, the emergence of pests and diseases (*Ren et al., 2012*), and soil moisture depletion (*Tao et al., 2003*). Conversely, enhanced photosynthesis, reduced drought-induced yield loss in the north, and increased multiple cropping index have presented potential avenues for improvement. The potential of wheat production can be realised through the implementation of various strategies, including the breeding of new varieties (heat tolerance, drought resistance, disease resistance, *etc.*) and the optimisation of farmland management practices (precision agriculture, water-saving irrigation, mulch planting, *etc.*).

## Variety breeding

The breeding characteristics of Chinese wheat are heat tolerance, drought resistance and disease resistance (Table 2). For heat tolerance, physiological and growth-development aspects are regarded as breeding goals respectively. For example, an isoleucine residue in the wheat Rubisco-activating enzyme functions as a heat-and-regulation switch, which can maintain photosynthetic efficiency at high temperatures (*Degen, Worrall & Carmo, 2020*);

drawing on desert plants to adjust the ratio of membrane lipids to reduce membrane damage, *etc.* (*Prasertthai et al., 2022*). For drought resistance, analysis, water management, and growth regulation are regarded as breeding goals respectively, such as having deep and extensive root systems, efficiently regulating osmotic pressure, strengthening hormone balance, and regulating stomatal characteristics, *etc.* For disease resistance, specific diseases and comprehensive resistance are regarded as crop breeding goals respectively. For example, varieties with aggregated Yr5, Yr15 and Yr18 rust-resistance genes can defend against rust (*Li et al., 2016*); varieties with introduced Pm genes have enhanced resistance to powdery mildew, *etc.* (*Wang et al., 2023a*). It is also possible to carry out the selection and breeding of multi-resistant varieties to ensure the wheat yield in China.

Breeding methods for developing climate-adaptable wheat varieties include traditional cross-breeding, mutation breeding, molecular-marker-assisted breeding and genetic-engineering breeding (*Selva et al., 2020*; *Liu et al., 2021b*; *Song et al., 2023*). Traditional cross-breeding can integrate genetic diversity, is simple to operate and has mature experience, but it has a long breeding cycle and is restricted by gene linkage and the accuracy of phenotypic selection (*Selva et al., 2020*); mutation breeding can create new variations and perform directional screening, but the mutations are random and mostly harmful, it is difficult to control the mutation frequency and type, and there are also safety issues (*Liu et al., 2021b*); molecular-marker-assisted breeding has accurate early-stage selection, is not affected by the environment and has high gene-pyramiding efficiency, however, the marker-gene linkage needs to be tight, the cost of marker development is high, and its effect on complex traits is limited (*Song et al., 2023*); genetic-engineering breeding can break species boundaries, precisely improve traits and is relatively fast, but it faces problems such as low public acceptance, unstable gene expression and high technical and intellectual-property-right thresholds (*Song et al., 2023*). Therefore, to achieve the goal of efficient breeding, multiple breeding methods should be comprehensively considered and flexibly applied according to local conditions.

Different wheat-growing areas in China can adopt a variety of breeding methods to cope with climate change. In the spring wheat-growing area of Northeast China, according to the characteristics of a short frost-free period and early occurrence of autumn early-frost, traditional cross-breeding and cold-tolerance gene mining are adopted. It is necessary to select and breed early-maturing varieties with better adaptation to low light intensity on the basis of high-temperature tolerance (*Ye et al., 2020*; *Selva et al., 2020*). In the spring wheat-growing areas of North and Northwest China, due to the characteristic of less precipitation, multi-gene pyramiding breeding (with the help of molecular-marker-assisted breeding), mutation breeding and directional screening are adopted. On the premise of heat resistance, it is necessary to select and breed drought-tolerant and high-water-use-efficiency varieties (*Sun et al., 2018*; *Liu et al., 2021b*). The winter wheat-growing area of North China, which is prone to dry-hot wind and pests and diseases, mutation breeding combined with traditional screening and multi-gene pyramiding breeding (combined with molecular-marker-assisted breeding) are adopted. It is necessary to select and breed multi-resistant and lodging-resistant varieties (*Zhang et al., 2023*; *Selva et al., 2020*; *Liu et al., 2021b*). The Middle and Lower Reaches of the Yangtze River winter wheat

**Table 2 Breeding goals of Chinese wheat in response to climate change.**

| Characteristics | Breeding goals | |
|---|---|---|
| Heat tolerance (*Mondal et al., 2021*) | Physiological aspect | Maintain photosynthetic efficiency at high temperatures, reduce membrane damage, and improve transpiration regulation (*Degen, Worrall & Carmo, 2020*; *Prasertthai et al., 2022*) |
| | Growth and development aspect | Ensure normal reproductive development at high temperatures, shorten the growth period sensitive to high temperatures (*Liu et al., 2020*) |
| Drought resistance (*Mondal et al., 2021*) | Root system | Developed root system, high-efficiency water absorption (*Akman, Yildirim & Bağci, 2023*) |
| | Water management | Leaf water retention, high water-use efficiency (*Liu et al., 2018*) |
| | Growth regulation | Have a drought-tolerant growth-hormone regulation mechanism (*Selva et al., 2020*) |
| Disease resistance (*Mondal et al., 2021*) | Specific diseases | Target rust, powdery mildew, fusarium head blight, *etc.* (*Li et al., 2016*; *Wang et al., 2023b*) |
| | Comprehensive resistance | Pursue durable and broad-spectrum disease resistance (*Nie et al., 2019*; *Liu et al., 2021a*) |

area, molecular-marker-assisted breeding is used to optimize disease resistance and moisture tolerance, and traditional cross-breeding and utilization of local variety resources are carried out. It is necessary to select and breed heat-tolerant and short-growth-period varieties to avoid the adverse effects of high temperature and high humidity on wheat grain filling (*Nie et al., 2019*; *Liu et al., 2021a*, *2021b*). The Southwest wheat area with complex terrain, multi-gene pyramiding breeding (with the help of molecular-marker-assisted breeding) combined with traditional cross-breeding of excellent genes from wild relatives is adopted. It is necessary to select and breed heat-resistant varieties adapted to different altitude differences; in the South China winter wheat area with abundant heat and frequent pests and diseases, mutation breeding combined with rapid screening technology and genetic-engineering breeding to introduce heat-resistant and disease-resistant genes are adopted. It is necessary to select and breed heat-resistant, moisture-tolerant, pest- and disease-resistant varieties (*Sun et al., 2018*; *Chen et al., 2018*; *Liu et al., 2021b*; *Song et al., 2023*). Special regions can also have certain tendencies. For example, 80% of the wheat fields in South Xinjiang have fruit trees, and shade-tolerant varieties need to be selected (*Li et al., 2019*). In the south, the proportion of non-agricultural water use has increased, and water-saving varieties need to be selected (*Xiong et al., 2010*). Although variety breeding can cope with climate change and ensure stable yield, the breeding time of wheat varieties is relatively long. Therefore shortening the breeding time of wheat varieties is the key to coping with the climate.

Genotype-environment interaction and multi-location trials are helpful for shortening the wheat breeding time (*Yue et al., 2022*; *Bischoff et al., 2022*). Genotype-environment interaction can achieve early-stage precise screening, optimize parent selection and directionally improve target traits, avoid later-stage operations on non-adaptive genotypes, reduce blind hybridization and precisely improve varieties, thus saving time (*Yue et al., 2022*). Multi-location trials can quickly evaluate adaptability, accelerate trait screening and predict variety performance in advance. Through multi-environment parallel screening, comprehensive and accurate screening of excellent genotypes and the establishment of

prediction models, unnecessary breeding generations and trial time can be reduced, thereby accelerating the breeding process (*Bischoff et al., 2022*). Moreover, the combination of genotype-environment interaction and multi-location trials can accurately evaluate the adaptability and stability of varieties and cultivate excellent varieties suitable for the climate-change environment. For example, the wheat genetic breeding team in the Crop Research Institute of Shandong Academy of Agricultural Sciences, combined with genotype-environment interaction through traditional cross-breeding, selected and bred "Jimai 22." And on the basis of "Jimai 22," the team in the Institute of Crop Sciences, Chinese Academy of Agricultural Sciences, adopted the molecular-marker-assisted method to select and breed "Jimai 23." During the breeding process of these two varieties, they were all evaluated and verified through multi-location trials to shorten the breeding time (*Jia et al., 2020*). Among them, "Jimai 22" won the second-prize of the National Science and Technology Progress Award in China. Therefore, adopting a variety of breeding methods in line with local conditions and combining them with genotype-environment interaction and multi-location trials is helpful to accelerate the breeding time of new varieties.

## Adjustment of farmland management measures

The cultivation and management technology adapted to local conditions can regulate the growth process of wheat to a certain extent based on meteorological warnings (*Bachmair et al., 2018*), and give full play to the potential of varieties, achieving the purpose of stable and increased yield (*Henriksen et al., 2018*). Biological control technologies such as birds, bees, butterflies, and spiders can not only increase farmland biodiversity but also reduce the impact of pests and diseases on crops and improve the stability of yield (*Shuqin & Fang, 2018*; *Barratt et al., 2018*; *Redlich, Martin & Steffan-Dewenter, 2018*). Furthermore, 55% of biological control methods are effective in controlling target weeds (*Schwarzländer et al., 2018*). The spring wheat-growing area of Northeast China has the capacity to adjust the sowing and harvest time in order to fully utilise the relatively high temperature in summer. Furthermore, the region is promoting dryland farming technology with the objective of improving water use efficiency (*Ye et al., 2020*). In spring wheat-growing areas of North and Northwest China, the planting structure is adjusted, and the types of crops rotated with wheat, such as soybeans and potatoes, are increased. This is done to maintain soil fertility and enhance the soil's ability to retain fertilizer and water (*Sun et al., 2018*). Additionally, dryland water storage and moisture conservation technologies, such as straw mulching, deep plowing and deep loosening, are adopted. Phosphorus fertilizer is also used to promote the development of wheat roots, thereby improving water use efficiency. Furthermore, the implementation of reasonable close planting has been shown to enhance the utilisation rate of light energy by the population (*Nie et al., 2019*). The winter wheat-growing area of North China employs a range of techniques to enhance ventilation management in wheat fields. These include reasonable close planting, intertillage and weeding, suppression, and spraying growth regulators with the objective of reducing the adverse effects of high temperatures. Additionally, the region utilises high-light-efficiency cultivation techniques, such as wide-narrow row planting and the use of reflective films,

with the aim of increasing the light-receiving area of plants (*Zhang et al., 2023*). The winter wheat-growing area of the middle and lower reaches of the Yangtze River demonstrates a commitment to the judicious application of nitrogen, with the objective of preventing excessive plant growth. Additionally, the region employs a systematic approach to pruning, with the aim of reducing plant redundancy and enhancing light conditions between plants (*Liu et al., 2021b*). The winter wheat-growing area of Southwest China employs reasonable intercropping and relay cropping techniques to enhance multiple cropping and the utilisation rate of annual light radiation (*Ye et al., 2020*). The winter wheat-growing area of South China optimises planting density to prevent poor ventilation caused by an overly large population 84. And all wheat-growing areas in China must reasonably adjust the amount of fertilization, especially nitrogen fertilizer, to fully utilize the increased $CO_2$ to promote photosynthesis. It is important to note that in the northern wheat-growing regions, the prevalence of pests and diseases, as well as lodging caused by heavy rainfall, must also be taken into account. In the southern wheat-growing areas, there is a need to investigate new cultivation techniques in relation to drainage, reducing plant redundancy, and the physiological drought of plants caused by high temperatures (*Sun et al., 2018*; *Nie et al., 2019*). Moreover, for biological control, new species should be scientifically introduced to reduce the risk of biological invasion (*Heimpel & Cock 2018*).

Moreover, the combination of wheat field management practices with variety selection and breeding can further improve the climate adaptability of wheat fields. Wheat-growing areas should select varieties according to their own climatic characteristics and support corresponding management measures, and at the same time adjust management strategies according to the growth characteristics of varieties (*Chen et al., 2018*; *Zhu & Song, 2021*; *Song et al., 2023*). For example, in the arid Northwest spring wheat area, varieties with strong drought tolerance should be selected, such as those with developed root systems and high-degree leaf cutinization, and in wheat field management, more attention should be paid to soil moisture conservation, using minimum-tillage and no-tillage techniques, and appropriately increasing the sowing depth during sowing (*Ye et al., 2020*; *Liu et al., 2021a*). Meanwhile, when determining target traits in breeding, the needs of wheat field management should be fully considered. For example, when breeding in the Northeast spring wheat area, varieties with moderate plant height, tough stems that are not easy to lodge, and suitable growth periods should be cultivated, so as to facilitate mechanized harvesting and deal with the situation of a short growth season (*Ye et al., 2020*; *Selva et al., 2020*). Moreover, the feedback from wheat field management can promote breeding improvement. For example, in the southern wheat-growing areas such as the Middle and Lower Reaches of the Yangtze River winter wheat area, it has been found that the resistance of pest- and disease-resistant varieties is insufficient. This feedback to breeders promotes the adjustment of breeding directions, strengthens the research on new disease-resistant genes or resistance mechanisms, and cultivates varieties more adapted to the new environment (*Nie et al., 2019*; *Liu et al., 2021a*, *2021b*). In addition, a comprehensive regional adaptability plan should be established, including variety selection, breeding goals and wheat field management practices, and the comprehensive strategy should be

dynamically adjusted according to climate change, with all parties closely cooperating to meet the challenges.

Mulching in farmland management measures can significantly adjust the temperature of wheat fields and soil moisture, which plays an important role in China's wheat response to rising temperatures and spatio-temporal non-uniformity of precipitation in the context of climate change (*Ye et al., 2020*; *Nie et al., 2019*). In terms of mulching materials, for example, using straw mulching in northern wheat-growing areas can effectively reduce soil moisture evaporation and play a role in soil moisture conservation, which has a significant effect in dealing with drought (*Ye et al., 2020*); while using plastic film mulching in the Northwest Spring Wheat Area has the functions of increasing temperature, retaining moisture and inhibiting weed growth (*Nie et al., 2019*). From the perspective of mulching time, winter mulching can increase soil temperature and reduce the impact of freezing damage on wheat. For example, in the Northeast spring wheat area and the North China winter wheat area, appropriate winter mulching helps wheat to survive the winter safely; mulching in the early growth stage of wheat is conducive to protecting seedlings and promoting the growth of young seedlings, and mulching in the later growth stage helps to maintain soil moisture and nutrients and increase yield (*Nie et al., 2019*; *Zhang et al., 2023*). In southern wheat-growing areas such as the Middle and Lower Reaches of the Yangtze River winter wheat area, mulching needs to be carried out with caution. The relatively closed and high-humidity micro-environment formed under the mulch provides more suitable living and breeding conditions for pathogens such as *Fusarium graminearum* of wheat, and the mulch will attract pests such as aphids to gather, increasing the difficulty and cost of pest and disease control (*Ye et al., 2020*; *Liu et al., 2021a*, *2020*). Therefore, rationally selecting mulching materials and mulching time and carrying out mulching measures in each wheat-growing area according to local conditions are helpful to improve the ability of wheat to cope with climate change.

## Farmland regional ecological construction

The regional ecology of farmland is capable of accommodating climate change and constitutes a crucial foundation for the realization of sustainable agricultural development. This encompasses a range of strategies, including the establishment of farmland shelterbelts, the creation of ecological ditches, soil improvement and protection, the formation of ecological field ridges, and other measures. The erection of farmland shelterbelts has been demonstrated to reduce wind speed at a height of 2 m above the ground by between 10% and 20% (*Ujah & Adeoye, 1984*). Similarly, the construction of ecological ditches has been shown to enhance the capacity of the soil to absorb and store precipitation, thereby reducing soil evaporation and increasing soil moisture availability (*Zhu & Song, 2021*). This, in turn, has been found to mitigate the adverse effects of drought on wheat (*Ye et al., 2020*; *Chen et al., 2018*), thereby creating conditions conducive to the collection of water for agricultural use and precision irrigation in the northern wheat-growing regions. Additionally, it provides convenience for drainage in the southern wheat-growing areas, while simultaneously reducing the incidence of pests and diseases caused by high temperatures and high humidity. Furthermore, shelterbelts have the

potential to absorb $CO_2$ and contribute to the mitigation of climate change (*Perry et al., 2012*). It promotes the decomposition of soil organic matter, increases the organic matter content in farmland, especially the increase in polysaccharides is relatively high, and is conducive to the restoration of soil fertility (*Dhillon et al., 2017*). Ecological field ridges have ecological functions such as planting slope protection plants and increasing the ecological landscape of farmland. It can be observed that the construction of farmland regional ecology can enhance the resilience and sustainability of farmland, which is of great significance for coping with changes in precipitation and temperature and enhancing the defensiveness of wheat fields against meteorological disasters.

Moreover, considering that China is a large-population country and water resources are relatively scarce, water-saving irrigation should be fully considered in farmland ecological construction (*Zhu & Song, 2021*). From the perspective of wheat-growing areas, in arid and semi-arid wheat-growing areas such as the Northwest spring wheat area, water-saving irrigation can effectively relieve the pressure of water resource shortage and ensure the water demand for wheat growth (*Nie et al., 2019*). The North China winter wheat area is faced with the problems of seasonal drought and uneven precipitation, and water-saving irrigation can improve the utilization efficiency of water resources (*Zhang et al., 2023*). Meanwhile, water-saving irrigation can be combined with farmland mulching. Mulching can enhance the soil water-holding capacity and reduce soil water evapotranspiration. For example, drip irrigation combined with ground mulch can reduce soil water evaporation (*Ye et al., 2020*; *Nie et al., 2019*). In addition, mulching in the early growth stage of wheat is helpful to ensure the temperature for seed germination and seedling growth, and mulching in the later growth stage of wheat is helpful to reduce the scouring of soil by rainwater, prevent the loss of soil nutrients, and at the same time adjust soil temperature and humidity and reduce the impact of waterlogging (*Ye et al., 2020*; *Zhang et al., 2023*; *Liu et al., 2021a*, *2020*) Therefore, through the synergy with mulching, water-saving irrigation improves the ability of wheat to adapt to climate change in different wheat-growing areas and ensures the stability of wheat yield and quality.

**Precision agriculture**

The climate monitoring and early warning system in precision agriculture is constantly developing and adopting new technologies and methods, which is an important measure to cope with climate change (*van Ginkel & Biradar, 2021*). The establishment of single monitoring and early warning systems for drought, high temperature and frost can improve the accuracy of the system (*Howe & Naumova, 2022*). In addition, satellite remote sensing and geographic information systems can provide real-time feedback on areas affected by crop disasters and recovery conditions (*Adedeji et al., 2020*). In the northern wheat-growing regions, the implementation of soil moisture and temperature monitoring systems can be enhanced to facilitate the timely irrigation of crops during periods of drought, as well as the selection of optimal planting times and crop varieties. In the southern wheat-growing zones, the establishment of a sophisticated precipitation monitoring network and a robust meteorological early warning system is crucial to enable proactive responses to flood control, drainage, and the mitigation of hot dry winds. With

the increase in temperature and $CO_2$ concentration, the monitoring of soil nutrients and pests and diseases should be strengthened to ensure the sustainable development of the soil and the prevention and control of pests and diseases. Therefore, the climate monitoring and early-warning system in precision agriculture is helpful to ensure the wheat yield.

Moreover, precision agriculture can also promote the improvement of wheat grain quality and the sustainable development of agriculture. With the help of satellite positioning systems, geographic information systems, remote sensing technologies and sensor technologies, precision agriculture can accurately obtain information about the growth environment and growth status of wheat (*Adedeji et al., 2020*). In terms of improving wheat quality, precision fertilization can precisely and variably apply fertilizers such as nitrogen, phosphorus and potassium according to the soil nutrient status and the growth stage of wheat, ensuring the balanced nutrition of wheat and improving quality indicators such as protein content; precision irrigation is carried out in accordance with the soil humidity and the water requirement law of wheat, avoiding the influence of excessive or insufficient water on wheat quality (*Chen et al., 2022*). In terms of environmental protection, precision agriculture reduces the excessive use of fertilizers and pesticides. Precision fertilization avoids fertilizer waste and reduces the risk of nutrient loss in the soil and eutrophication of water bodies; precision pest and disease control, through precise monitoring of the occurrence of pests and diseases, precisely applies pesticides only in necessary areas, reducing the pollution of soil, water bodies and air by pesticide residues, thereby improving wheat quality while achieving environmental protection and enhancing the sustainability of wheat production in response to climate change (*Li et al., 2016*).

# DISCUSSION AND PROSPECTS

## Discussion

The impact of climate change on wheat in China is characterised by a duality of effects, both favourable and unfavourable.

(1) The increase in temperature prolongs the reproductive stage of wheat (*Xiao, Bai & Liu, 2018*). It is expected to be prolonged by 1.5 days in 2050 (*He et al., 2015*). The rising temperature and the increased $CO_2$ concentration (*Gray, 2007*; *Valone, 2021*) work together to enhance photosynthesis and grain filling, increase grain weight (*Soliman, Shalabi & Campbell, 1994*), and thereby increase the yield per unit area. However, the prolonged reproductive stage is also more susceptible to the stress of hot, dry winds, which will also result in an increase in potential evaporation (*Chen et al., 2016*), thereby reducing the utilisation degree of soil moisture. This is particularly problematic in arid or semi-arid regions of the northern wheat-producing areas, where it can readily result in yield declines (*Ju et al., 2013*).

(2) Global precipitation is expected to increase significantly in the future. The precipitation increase is expected to exceed 75% from 2050 to 2080 (*Xiao, Bai & Liu, 2018*). The yield per unit area in the northern wheat-growing areas is expected to increase by 3.2–4.1% (*Geng et al., 2019*). In the southern wheat-growing areas, due to the high non-agricultural water consumption, the water consumption of wheat fields is reduced and the water use efficiency is improved (*Xiao, Bai & Liu, 2018*). However, the spatiotemporal

heterogeneity of precipitation in the future will further intensify (*Peterson et al., 2008*; *Gray, 2007*; *Sun et al., 2018*; *Song et al., 2019*; *Nie et al., 2019*), increasing the incidence of drought in the northern wheat-growing areas and flood disasters in the southern wheat-growing areas (*Ren et al., 2012*). Moreover, continuous rainy weather leads to an increase in pests and diseases, causing yield losses (*Xiong et al., 2010*).

(3) The increase in $CO_2$ concentration and temperature prompts the selection of early-maturing varieties in the northern wheat-growing areas, reducing the competition between wheat and multiple crops such as soybeans and corn (*Abdalla et al., 2020*). This has resulted in an increase in the multiple cropping index and expansion of the area of multiple cropping (*Hao et al., 2019*). For example, the cropping system in North China has expanded to two crops of wheat-corn, two crops of wheat-soybean, or 2 years and three crops of spring corn-wheat-millet. The total output is increased by increasing the planting area (*Long et al., 2010*). However, multiple cropping will increase the types of weeds (*Altieri et al., 2015*), which may cause a rapid decline in soil fertility (*Ramankutty et al., 2002*). Extreme dry weather will also cause the depletion of soil moisture (*Tao et al., 2003*), which is not conducive to the sustainability of the soil.

(4) Furthermore, climate change has resulted in a reduction in the cultivated area of wheat and a westward shift in the planting center. The beneficial consequence of this relocation is that, due to the rise in temperature, $CO_2$ concentration, and precipitation, in addition to the enhancement of farmland irrigation techniques, the wheat planting area in mountainous regions and arid lands has increased, and some western areas have initiated intercropping and multiple cropping of wheat. The unfavourable aspect is the reduction in the planted area, which is attributable to a confluence of factors, including rural labour migration, rising input costs, policy adjustments, land transfer, and climate change. This necessitates a more comprehensive investigation and examination of the influence of each factor on the planting area.

In conclusion, the future climate has the potential to increase domestic wheat production (*Xiao, Bai & Liu, 2018*), but it also faces challenges such as the rapid loss of soil nutrients and the intensification of pests and diseases (*Singh & Sidhu, 2014*). Furthermore, in the northern wheat-producing regions, the objective is to achieve high and super high yields as production increases. In contrast, in the southern wheat-producing regions, the goal is to attain stable and high yields while reducing production.

## Prospects

### A comprehensive investigation into the microscopic mechanisms underlying climate change and wheat growth

The majority of research on the impact of climate change on wheat has concentrated on indicators such as yield and phenology. However, there is a paucity of studies examining the response of wheat growth and development and yield formation to climate change. The microscopic mechanisms of wheat, including cell division and elongation, photosynthesis, and respiration, are particularly susceptible to environmental changes and operate with greater efficiency. A deeper understanding of the microscopic mechanisms of wheat can elucidate the mechanisms by which wheat adapts to climate change (*Zenda et al., 2021*).

This, in turn, can provide a theoretical basis for the development of new wheat varieties, the innovation of new agricultural technologies and methods, the mitigation of the adverse effects of climate change, and the assurance of food security.

### Strengthen the research on the comprehensive influence of multiple factors

While studies on single factors, such as temperature, precipitation, and $CO_2$ concentration, can yield fundamental insights into wheat yield performance, they are insufficient for accurately reflecting the complex and dynamic interactions between these factors in real-world conditions. The incorporation of multiple factors allows for a more accurate reflection of the environmental conditions and an explanation of the interactions among the factors, which is conducive to a comprehensive assessment of the adaptability strategies of wheat.

### Utilize new monitoring and simulation technologies

The precision and accuracy of new monitoring and modelling technologies are superior, enabling comprehensive depiction of subtle changes in wheat through integration of meteorological, soil, and wheat indicators (*Ma et al., 2022*). These technologies facilitate more realistic spatiotemporal dynamic simulations of climate and wheat growth and development at varying scales, assess the uncertainty of input parameters, and enhance the reliability and credibility of research results (*Zare et al., 2022*). Furthermore, they are more likely to integrate with big data analysis, artificial intelligence, and machine learning, thereby augmenting prediction and simulation capabilities. This facilitates the reflection of the adaptive changes of wheat to the climate, thereby enabling the formulation of agricultural policies that are responsive to climate change.

### Funding

This research was funded by the National Key R&D Program of China (No. 2021YFD1901102) and National Wheat Industry Technology System Special Project (No. CARS-03-01-24). The funders had no role in study design, data collection and analysis, decision to publish, or preparation of the manuscript.

### Grant Disclosures

The following grant information was disclosed by the authors:
National Key R&D Program of China: 2021YFD1901102.
National Wheat Industry Technology System Special Project: CARS-03-01-24.

### Competing Interests

The authors declare that they have no competing interests.

## Author Contributions

- Yu-chen Fan conceived and designed the experiments, performed the experiments, analyzed the data, prepared figures and/or tables, authored or reviewed drafts of the article, and approved the final draft.
- Ya-qi Yuan conceived and designed the experiments, performed the experiments, analyzed the data, prepared figures and/or tables, authored or reviewed drafts of the article, and approved the final draft.
- Ya-chao Yuan conceived and designed the experiments, performed the experiments, analyzed the data, prepared figures and/or tables, authored or reviewed drafts of the article, and approved the final draft.
- Wen-jing Duan conceived and designed the experiments, performed the experiments, analyzed the data, prepared figures and/or tables, authored or reviewed drafts of the article, and approved the final draft.
- Zhi-qiang Gao conceived and designed the experiments, performed the experiments, analyzed the data, prepared figures and/or tables, authored or reviewed drafts of the article, and approved the final draft.

## Data Availability

This is a literature review.

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
