# Peer review of "Research progress on the impact of climate change on wheat production in China"

_PeerJ, doi:10.7717/peerj.18569_

## Round 0.1 · original submission · Major Revisions

Thank you for submitting your manuscript entitled "Research Progress on the Impact of Climate Change on Wheat Production in China]" to PeerJ. After carefully considering the reviews provided by two reviewers, we have decided on your submission.

Reviewer 1 has raised concerns about the manuscript clarity, noting that the content appears to lack a thorough discussion of the topic. He highlighted a significant deficiency in personal analysis and the interconnection between different subsections, ultimately recommending rejection of the manuscript in its current form.

Reviewer 2, however, believes that substantial revisions could improve the manuscript significantly.

Based on these evaluations, we believe that your manuscript has the potential to make a valuable contribution to the field. Still, comprehensive revision is required to meet the standards of PeerJ. Therefore, we are requesting a major revision of your manuscript. Please address the following points thoroughly:

Clarity and Depth: Enhance the manuscript clarity by elaborating on key points and providing a more in-depth discussion on the topic. Ensure each section is well-developed and interconnected, demonstrating a clear progression of ideas.

Personal Analysis: Integrate a more personal analysis throughout the manuscript. Provide critical insights that reflect your unique perspective and expertise on the subject matter.

Interconnection of Subsections: Ensure a cohesive flow between different manuscript sections. Each subsection should logically build upon the previous one, contributing to a comprehensive understanding of the topic.

Specific Reviewer Comments: Carefully address all particular comments and suggestions made by both reviewers. Include a detailed response to each point raised, indicating how you have revised the manuscript accordingly.

We believe that with these substantial revisions, your manuscript can meet the high standards required for publication in PeerJ. We look forward to receiving your improved manuscript.

Thank you for your understanding and cooperation.
Sincerely,
Elsayed Mansour

·

Basic reporting

Report Analysis: Research Progress on the Impact of Climate Change on Wheat Production in China
Aim: This article examines the impact of climate change on wheat production in China.
Suitability of experimental design: This is a review article, not an original research study. Therefore, it doesn't have an experimental design.
Key findings:
• Climate change is causing a rise in temperature, increased precipitation variability, and higher CO2 concentration.
• These changes are affecting wheat phenology, with a shortened growing season and a shift in the centre of cultivation westward.
• While some regions might see yield increases due to these changes, others might experience yield reduction.
• Climate change is also increasing the risk of agricultural disasters like drought and pest infestation.
Strengths:
• Addresses a timely and important topic: Climate change and its impact on food security is a major global concern. This article focuses on a specific crop (wheat) in a specific region (China), making the research valuable.
• Comprehensive Literature Review: The methodology section highlights a clear search strategy, indicating the authors reviewed relevant research.
• Balanced Viewpoint: The article acknowledges both potential benefits (increased yield in some regions) and drawbacks (decreased yield in others) of climate change.
• Clear Structure and Writing: The logical flow and concise writing make the information easy to understand.
Weaknesses:
• As a review article, it doesn't present new data.
The focus is mainly on the negative impacts of climate change
Weaknesses/ Areas for Improvement (Minor Revisions):
• Deeper Exploration of Positive Effects: While acknowledging potential yield increases, the article could delve deeper into the specific regions and conditions that might benefit. This would provide a more nuanced perspective.
• Inclusion of Adaptation Strategies: The conclusion mentions the need for adaptation strategies, but briefly. Adding a section or including more references on existing or potential adaptation methods (e.g., drought-resistant wheat varieties) would strengthen the article's practical value.

The writing needs a little modification; the authors seem to use difficult/uncommon synonyms to refine the sentences of the original texts to reduce the plagiarism or so which is not bad but make sure to use the appropriate terms. I have some minor comments to the authors to, some can be fixed, the others would require adequate justification. I’ve embedded them into the main document
Overall, this is a well-researched and informative piece that contributes to the understanding of climate change's impact on wheat production in China. With the suggested revisions, the article can provide even greater value to readers and researchers in the field.

Experimental design

The authors describe their methodology for selecting the literature they reviewed. They used keywords related to climate change, wheat, and China to search academic databases.

Validity of the findings

The findings are based on a review of existing literature, so their validity depends on the quality of the studies reviewed. The authors appear to have used credible sources like peer-reviewed journals

Reviewer 2 ·

Basic reporting

The review paper "Research progress on the impact of climate change on wheat production in China",
which tried to discusses the effect of climate change on wheat production in china.

Overall, the paper needs a profound English editing, with improved sentences and english fluidity. A lack
of clarity was clearly observed when reading the paper. Morevover, the paper content was mostly
superficial and the topic was not discussed thoroughly with a clear lack of personal discussion and ineterconnection between different subsection.

Therfore, i recommende the re-submission of the paper after a profound revision.

Experimental design

'no comment'

Validity of the findings

'no comment'

Additional comments

'no comment'

---

## Round 0.2 · Minor Revisions

Dear Authors,

Thank you for submitting the revised version of your manuscript. The revised manuscript has been reviewed by the initial two reviewers. Reviewer 1 has accepted the modifications you made, acknowledging that the changes addressed their concerns satisfactorily. However, Reviewer 2 was not convinced by the revisions, citing a lack of substantial improvement compared to the original submission, as a result they opted for Rejection.

To ensure a fair evaluation, I sent the manuscript to a third reviewer for an additional perspective. The third reviewer has recommended revisions, highlighting specific areas that need further improvement.

Based on this feedback, I recommend that you revise your manuscript according to the comments and suggestions provided by the third reviewer. Please carefully address all the points raised and submit the revised version for further consideration.

We look forward to your revised submission.

·

Basic reporting

The authors made a substantial improvement in the manuscript and took good care of the appointed comments. I believe that the manuscript is now ready for publication in PeerJ

Experimental design

N/A

Validity of the findings

N/A

Reviewer 2 ·

Basic reporting

Dear authors,

I re-revised the review paper "Research progress on the impact of climate change on wheat production in China".

Unfortunatly, althought the change that you have done, but there is no significant improvement in the paper compared to the last version.

In my opinion, the paper still not able to be published in the PeerJ journal.

Experimental design

'no comment'

Validity of the findings

'no comment'

Additional comments

'no comment'

Reviewer 3 ·

Basic reporting

No comments

Experimental design

No comments

Validity of the findings

No comments

Additional comments

The paper provides a comprehensive overview of the complex interplay between climate change and wheat production in China. It outlines the current climate change scenario, its direct and indirect impacts on wheat cultivation, and potential adaptation strategies.
-Comments and Suggestions for Authors
- Abstract
- In American English, "center" is typically used, while "centre" is more common in British English. Please unify and use the most common word in all manuscript.
-Consider adding more specific details or examples to illustrate the impacts of climate change on wheat production. For instance, you could mention the extent of yield reductions or the specific regions affected.
- Introduction
- Overall, the introduction provides a strong foundation for the research. It effectively establishes the context of climate change and its impacts on wheat production in China.
-While the introduction provides general information about climate change impacts, incorporating specific examples or case studies could strengthen the argument and make it more relatable.
-Consider citing additional relevant literature to further support the claims made in the introduction.
-The transition from discussing climate change impacts to introducing wheat production could be made smoother with a clearer connecting sentence.
- Survey Methodology
- The methodology could provide more details on the specific criteria used to select articles for inclusion in the review, such as citation analysis or relevance to the research questions.
- While the methodology mentions the use of crop models, it would be helpful to provide more information on the specific analysis techniques used to assess the impact of climate change on wheat production.
- Acknowledging potential limitations of the research, such as the possibility of missing relevant studies or the challenges of comparing data from different regions, could strengthen the methodology.
- In Section Impact on wheat phenology (lines 179-211), the text could be enhanced by discussing potential adaptation strategies for wheat producers to mitigate the negative impacts of climate change.
- In Section Effects on wheat grain yield (lines 236-277)
- Restructure the text to first discuss the general impact of each environmental factor, then delve into the variations between regions and wheat types.
- Include a table or chart summarizing the key points about yield changes in different regions and wheat types.
- Explain the biological mechanisms behind the observed responses.
-In section adaptive strategies and measures for chinese wheat in response to climate change:
-Provide more details on specific breeding objectives for new wheat varieties, such as heat tolerance, drought resistance, or disease resistance. Discuss potential management practices like precision agriculture, water-saving irrigation, or cover cropping.
-Highlight successful examples of adaptation strategies implemented in different regions of China. This could include farmers who have adopted new varieties or management practices to mitigate the effects of climate change.
-In Section Variety breeding:
- Discuss the importance of genotype-environment interactions and the use of multi-location trials to evaluate variety performance.
- Briefly mention the breeding methods used to develop climate-resilient varieties, highlighting the advantages and limitations of each approach.
- In Section Adjustment of Farmland Management Measures:
-Discuss how management practices can be integrated with other strategies, such as variety selection and breeding, to enhance climate resilience.

---

## Round 0.3 · accepted · Accept

Dear Authors,
Thank you for submitting the revised version of your manuscript. We appreciate the substantial improvements made to the manuscript. After careful evaluation of your revisions, it is clear that you have responded comprehensively to the reviewers' suggestions, significantly enhancing the clarity, rigor, and overall quality of the work.

I am pleased to inform you that it is now ready for publication. Congratulations on your excellent work, and thank you for your contribution to PeerJ.

Best regards,
Elsayed Mansour

Notes from the Section Editor:

some minor issues to clean up:

- line 786. I would think "Author Contributions" would be better than "Supplemental Material".
- line 789 "oe" -> "of".
- lines 758, 769, and 776. The first "sentence" of each of these paragraphs seem to be subheadings, and should be formatted as such.
- Table 2 should be reformatted so that words don't break (most apparent in column 2)

Reviewer 3 ·

Basic reporting

no comment

Experimental design

no comment

Validity of the findings

no comment

Additional comments

The authors have made the changes I suggested in the last review. I recommend its publication in this journal.